# Teach Large Language Models the Concept of Metacognition to Reduce Hallucination Text Generation

## Abstract

We introduce an algorithm that endows language models with enduring meta-cognitive capabilities. Inspired by meta-learning, our approach involves fine-tuning models on diverse datasets, including the original untuned base model. Throughout each training iteration, we randomly select various fine-tuned model versions, gauge their meta-cognitive capacities, and employ the meta-cognitive error average as the loss function for gradient updates. This empowers these models to assess their competence when interpreting human instructions, thereby averting the generation of responses beyond their abilities and mitigating hallucinatory text production. The meta-cognitive ability will be adapt to various fine-tuned versions of the main model, providing evaluations that align with the fine-tuned models' knowledge capacity.

## 1 Introduction

Since the end of 2022, with the release of ChatGPT by OpenAI, large language models (LLMs) has seen a dramatic improvement in the linguistic capabilities. This progress has sparked a swift and widespread adoption of large models, both in their development and application, sweeping across the global landscape.

While the remarkable capabilities of large language models (LLMs) are undeniable, the challenge of hallucination remains a persistent concern, exerting limitations on the advancement and implementation of these expansive models. It's worth noting that when we use the term "hallucination", we are not alluding to nonsensical, randomly strung together words akin to the musings of a drunk person or the hypothetical output of infinite monkeys typing away. In fact, the hallucinatory text generated by LLMs often adheres to correct grammar and exhibits coherent logic. However, it is plagued by a singular issue: a departure from established ground truth.

The origins of hallucination have yet to be comprehensively explored. As outlined in (Ji et al., 2023b), hallucinations can be categorized into two distinct types based on their sources: those stemming from data and those arising during training. Hallucinations originating from data are relatively straightforward to grasp: large-scale language models (LLMs) may have inadvertently internalized incorrect information from the provided data. Fortunately, there exists a straightforward approach to mitigating such instances: an increase in data cleaning efforts. Hallucinations originating from training are relatively more intricate to address.

In fact, hallucinations stemming from data often manifest as isolated pieces of information or factual errors, akin to an individual accidentally misremembering details. On the other hand, hallucinations arising from training frequently involve the elaborate fabrication of non-existent references or absurdly concocted extensive historical accounts—much like the work of a seasoned con artist. As a result, these types of hallucinations can significantly mislead users, potentially leading to societal harm and inducing a sense of panic.

The Reinforcement Learning from Human Feedback (RLHF) training algorithm (Ziegler et al., 2020) plays a pivotal role in endowing contemporary LLMs with their remarkable capabilities. Distinguishing itself from conventional fine-tuning, the innovation of RLHF lies in its utilization of human preference data to construct a reward model, subsequently employed in reinforcement learn-

ing applied to the LLM. As a result, the RLHF algorithm frequently enhances the model's alignment with human requirements. However, what are the shortcomings of RLHF?

While conclusive evidence may be lacking, based on intuition, the RLHF algorithm appears to incentivize language models to generate responses that are formally elaborate and professional, rather than candidly admitting their lack of understanding regarding a user's inquiry. In this context, LLMs resemble students confronted with a closed-book examination. Similar to how students often opt to provide an answer even if it's speculative, rather than leaving a question unanswered, LLMs may tend towards fabricating responses. This approach might yield a higher probability of yielding some points, with minimal perceived negative consequences. This is, in our opinion, the main source of hallucination of recent LLMs.

Giving negative score for fabricating seems to be an obvious solution. But, there are two reasons why this scenario underperforms. First, LLMs, including reward models used in RLHF algorithm, are statistical models. They are good at fitting the co-occurrence probability distribution of tokens, but this is not enough to fit the world's model. Second, it is easy to train a model that often say sorry and refuse to answer, but that kind of model is not very interesting and useful. It's challenging to acheive a balance between usefulness and reliability.

To furnish LLMs with an accurate depiction of real-world facts, enabling the model to scour the internet or reputable databases emerges as a viable approach (Nakano et al., 2021)(Varshney et al., 2023). However, this approach exhibits two distinct drawbacks.

Firstly, it can lead to inefficiency in cases where the model has already encoded relevant knowledge within its parameters. Given the expansive scale of LLMs, which can encompass hundreds of billions of parameters, users often necessitate multiple high-cost GPUs for inference. This process consumes a substantial amount of power, contributes to response latency, and results in a non-negligible carbon footprint, see (Luccioni et al., 2023)(Khowaja et al., 2023)(Fischer, 2023)(Rillig et al., 2023). Consequently, mitigating unnecessary searches would prove advantageous.

Secondly, the presence of low-quality content on the internet introduces the potential for the model to derive erroneous information, subsequently leading to the generation of hallucinatory text. Consequently, while the utilization of web searches offers a solution, it remains far from flawless.

Can we instill in LLMs the discretion to search only when absolutely necessary? This issue can be reframed as a task of teaching LLMs to accurately self-evaluate their own capabilities before generating text. In the realm of psychology, this ability, which is commonly possessed by most adults, is referred to as metacognition.

A straightforward approach involves creating a substantial set of instruction-response pairs and assigning scores to the generated responses. Through this process, LLMs can be trained to recognize instances where provided instructions that exceed their capabilities and make them more likely to generate hallucinatory responses. Subsequently, the models can take actions, such as conducting internet searches, to enhance the quality of their responses. However, this method presents an apparent drawback: the model tends to simply commit its current knowledge database to memory. Consequently, when additional knowledge is incorporated (e.g., via mechanisms like LoRA (Hu et al., 2022)), the metacognitive ability remains static. This necessitates fine-tuning the metacognition whenever the model's knowledge database is updated—a far from optimal situation.

In this paper, we present an algorithm that equips language models with metacognitive abilities that persist even after fine-tuning. Our algorithm draws inspiration from meta-learning. In simple terms, we acquire models fine-tuned on various datasets, including the original untuned base model. Subsequently, during each training iteration, we randomly select several versions of the fine-tuned models, compute their metacognitive abilities, and employ the average of the metacognitive errors as the loss function for gradient updates.

- Our primary contribution is the provision of an algorithm that imbues large language models with metacognitive capabilities, enabling these models to preemptively assess their ability to respond when presented with human instructions. This prevents the generation of answers when the model lacks the capability to do so, consequently reducing the occurrence of hallucinatory text generation.

- We also make another contribution by providing a new ground truth dataset comprising 16,000 questions across 160 finely segmented domains. This dataset can be employed for tasks like fine-tuning and evaluation of large language models.

## 2 RELATED WORK

### 2.1 LARGE LANGUAGE MODELS

The trend of large language models has emerged only within the past few years. In 2017, (Vaswani et al., 2017) introduced the Transformer architecture, which has proven to be highly effective in capturing long-range contextual relationships. It, alongside the pretraining-finetuning approach proposed in BERT (Devlin et al., 2018), forms the foundation of the majority of contemporary large language models.

Within the realm of large-scale language models, the decoder-only architecture stands out as one of the most active branches, well-suited for text generation. A standout example is the GPT series introduced by OpenAI, see (Radford & Narasimhan, 2018)(Radford et al., 2019)(Brown et al., 2020)(Ouyang et al., 2022)(OpenAI, 2022)(OpenAI, 2023). In the first three iterations of the GPT series, OpenAI's researchers trained models using extensively cleansed, large-scale internet text corpora. After that, they used human preference data to perform reinforcement learning. The model architecture utilized is a unidirectional attention decoder-only Transformer, with the training objective primarily focused on next-word prediction. However, both the model capacity and the volume of participating training data have witnessed substantial augmentation. Parameter counts have escalated from approximately 100 million to 1.5 billion, and further to a staggering 175 billion, while the textual data corpus has expanded from around 5GB Wikipedia and BookCorpus text to over 500GB cleaned out from 45TB text in Common Crawl. Upon scaling language models to the magnitude of GPT-3, researchers observed a non-linear growth in their capabilities, a phenomenon referred to as "emergence".

The success of ChatGPT has motivated more companies and independent researchers to develop other competitive large language models. Examples include Google's Bard (Google, 2023), Anthropic's Claude (Anthropic, 2023), Meta's LLaMA series (Touvron et al., 2023a)(Touvron et al., 2023b), Alpaca (Stanford University's self-instruct finetuned LLaMA) (Taori et al., 2023), Baidu's ERINE-bot (Baidu, 2023), Tsinghua University's GLM (Du et al., 2022)(Zeng et al., 2022), and independent developer Peng Bo's RWKV (based on a modified version of RNN rather than Transformer) (Peng et al., 2023).

### 2.2 HALLUCINATION PROBLEM IN LARGE LANGUAGE MODELS

Hallucination in large-scale language models (LLMs) has been a persistent concern since their inception. The issue of hallucination has been extensively explored across various text generation tasks, including summarization (Huang et al., 2023) and dialogue generation (Shuster et al., 2021). A comprehensive examination of the phenomenon of LLM hallucination across different sources and tasks was conducted in (Ji et al., 2023a), which also introduced certain metrics for the detection of hallucinatory outputs.

There are several existing hallucination detection datasets as benchmarks. As a benchmark, the work in (Liu et al., 2022) introduced a token-level reference-free hallucination detection dataset known as HADES. (Li et al., 2023) introduced another large-scale hallucination evaluation benchmark called HaluEval, which includes 35,000 hallucinated/normal samples for LLMs analysis and evaluation, by a ChatGPT-based two-step framework called sampling-then-filtering.

There are various methods in solving the hallucination problem.

One way to detect hallucinatory text involves utilizing uncertainty metrics available during the inference process of large models, such as token perplexity or entropy, as empirical evidence suggests that higher uncertainty is often associated with hallucinatory text generation. Both BARTScore (Yuan et al., 2021) and GPTScore (Fu et al., 2023) employ this approach. However, such methods require users to have access to internal data from the model's inference process, which is often unattainable in many scenarios, such as when using closed-source models like ChatGPT through API interfaces.

A distinct approach called SelfCheckGPT is presented in (Manakul et al., 2023). Unlike other methods, SelfCheckGPT solely relies on sampled responses and is applicable to black box models. It leverages the premise that if an LLM is well-informed about a particular subject, the responses it generates should demonstrate internal consistency and include logical facts. Conversely, when hallucinated facts are present, stochastically produced responses tend to diverge and conflict with each other. Consequently, in SelfCheckGPT's perspective, a greater diversity observed in the responses randomly drawn from the model corresponds to a heightened likelihood of these responses containing hallucinatory content.

Another approach is shown in (Feldman et al., 2023). By employing contextual information in tandem with embedded tags, the authors propose a strategy to effectively counter hallucinations in generative language models. To this end, they establish a baseline for hallucination frequency using prompt-response pairs devoid of context, using generated URLs as markers of potentially fabricated data. Notably, the inclusion of context alongside question prompts for the evaluated generative engines led to a noteworthy reduction in overall hallucination instances.

## 3 MODEL SELECTION AND DATASET PREPARATION

### 3.1 OVERVIEW

To embark on research concerning metacognitive capabilities, our initial step involves the selection of a model.

In this context, ChatGLM-6B (Zeng et al., 2022) emerges as a bilingual language model proficient in generating high-quality, multi-turn dialogues in both English and Chinese. Nonetheless, it stands relatively modest in size (6B parameters) compared to more extensive counterparts such as GPT-3 (175B parameters), PaLM (540B parameters), LLaMA (65B parameters), and the like. This distinction leads to a constrained reservoir of knowledge within the model. Consequently, situations involving finely-segmented and specialized domain inquiries often transcend the model's capacities, thereby heightening its susceptibility to hallucinations in such contexts.

Hence, to assess the enhancing impact of our approach on the model's meta-cognitive capabilities, we should prepare a broader array of specialized domain questions for the model. Additionally, it is imperative that these questions adhere to an objective factual nature (such as "What is the distance between the Earth and the Moon?") rather than leaning towards subjectively crafted and imaginative queries (like "Write an essay about the picturesque beauty of autumn").

Following this approach, we opted for 160 finely-segmented specialized domains in the fields of natural and social sciences as our subject areas. Utilizing GPT-3.5, we procured 100 distinct factual questions for each subject, accompanied by reference answers, culminating in an aggregate of 16,000 question-answer pairs. This comprehensive dataset serves as the foundation for training and evaluating ChatGLM-6B. Upon manual examination, GPT-3.5 demonstrates a commendable proficiency in accurately addressing factual questions posed to itself, with virtually no instances of generating hallucinatory text. We will call this dataset "ChatGPT ground truth" dataset. All question-answer pairs in this dataset are English, since GPT has best performance for English language.

Simultaneously, we also need ChatGLM-6B to generate answers for these 16,000 questions, which will be used for the subsequent evaluation.

To execute our algorithm, we also require a set of fine-tuned models built upon ChatGLM-6B using various knowledge bases. For flexibility and cost-effectiveness, these fine-tuned models will be trained using the LoRA technique. LoRA (Hu et al., 2022), which stands for Low Rank Adaption, is a technique that involves injecting a small number of trainable parameters into various layers of a model. This enables the creation of a customizable and combinable fine-tuning model.

Numerous options exist for the knowledge base of these LoRAs. As a preliminary exploration, we will opt for a straightforward approach: we will utilize 160 domains from the "ChatGPT ground truth" dataset to train 160 LoRA models. Each LoRA model will undergo training using a set of 100 questions associated with a single domain from the dataset. We emulate human metacognitive abil-

ities: after acquiring some knowledge in a certain domain, there is an increased sense of confidence when responding to specialized questions in that field.

## 3.2 GENERATE QUESTION-ANSWER PAIRS

We generate question-answer pairs by ChatGPT.

For each of the 8 large subjects: mathematics, physics, biology, chemistry, engineering, medicine, computer science and social science, we manually choose 20 finely-segmented specialized domains, as topics in generating question-answer pairs. Thus, we choose 160 topics in total. The full topic list is in Appendix A.

Then, for each topic, we will generate 100 factual questions by ChatGPT API. The initial prompt is simply "Generate {M} factual questions about {topic}: 1.", where M=20 is the number of factual questions to generate at once. Then, we will randomly choose 3 generated factual questions as examples. The prompt became:"Generate {M} factual questions about {topic}: 1.{first question} 2. {second question} 3. {third question} 4.". Then, we add generated questions, and delete the duplicated generations. We will repeat the generation and random choosing process until there are 100 non duplicated factual questions for a certain topic.

The next step is to generate answers for all the 16000 factual questions, also by ChatGPT. If we allow ChatGPT to directly answer questions, some queries may be challenging to answer accurately due to a lack of clarity regarding the topic. For example, the question "What is the definition of a submersion?" will have totally different answer when given the topic "topology". Thus, we will deploy the following prompt format: "Please answer a question about {topic} : {question}". For example, the question above will correspond to the prompt: "Please answer a question about topology : What is the definition of a submersion?" This prompt format effectively reduces ambiguity, enhancing the accuracy of responses.

## 3.3 TRAIN LoRAS FOR CHATGLM-6B

We simply apply the 100 question-answer pairs of each topic to train LoRA models for ChatGLM-6B. Since our purpose is model metacognition, not the knowledge itself, we will not separate train and test datasets in this step. Thus, we get 160 LoRAs of ChatGLM-6B, each is for one specific subject.

## 3.4 GENERATION AND EVALUATION OF CHATGLM-6B'S ANSWERS

For each of the 16000 questions, we generate ChatGLM-6B's answer, and ChatGLM-6B+LoRA(of the corresponding topic)'s answer. Then, we will use ChatGPT API to evaluate those answers.

By practice, we found that, for ChatGPT (at least 3.5 version) to give an accurate evaluation, the most important prompt technique is providing the grounding truth. If we directly let ChatGPT to evaluate ChatGLM-6B's answer, ChatGPT will believe almost all those answers are correct and clear. But if provided the ground truth (it is just the answer generated by ChatGPT itself), ChatGPT can detect hallucinations and provide a relatively accurate evaluation.

Thus, We have designed prompts for evaluation that utilize Markdown syntax, comprising four paragraphs connected sequentially: the question, ground truth, model-generated response, and evaluation. We ask ChatGPT to classify the answer into five types: A. clear answer; B. minor error; C. severe hallucination; D. nonsense; E. refuse to answer or other cases. The full prompt is provided in Appendix B.2.

Finally, the distribution of the dataset into five classes are shown in Table 1.

Table 1: Distribution of evaluation (A. clear answer; B. minor error; C. severe hallucination; D. nonsense; E. refuse to answer or other cases.)

| MODEL TYPE | A | B | C | D | E |
|---|---|---|---|---|---|
| original model | 2658 | 9795 | 2946 | 460 | 141 |
| model with domain specific LoRAs | 5750 | 9162 | 1029 | 56 | 3 |

Table 2: Evaluations and scores

| EVALUATION | SCORE |
|---|---|
| A. clear answer | 5 |
| B. minor error | 4 |
| C. severe hallucination | 2 |
| D. nonsense | 1 |
| E. refuse to answer or other cases | 3 |

## 4 META-COGNITION TRAINING

### 4.1 ALGORITHM

We will ask the model itself (using a specialized evaluation LoRA) to generate the evaluation, furnishing only the question. In line with our objectives, we must assess before responding, and therefore, the answer will not be provided when evaluating.

Our aim is to train the evaluation LoRA $E$ to adapt to various fine-tuned versions of the main model, such that it can provide evaluations of questions that align with the model's knowledge capacity. For the original model $M$, we denote by $M_i$ the $i$-th finetuned version of $M$. Also, denote $M_0 = M$. We write the evaluation of a question $q$ given by $M_i$ + evaluation LoRA $E$ as $\text{Evaluation}(M_i, E, q)$. Then, The training objective is to minimize the following loss function:

$$\text{Loss}(E) = \frac{1}{N} \sum_i w_i \sum_{j \in J_i} \text{Distance}(\text{Evaluation}(M_i, E, q_j), \text{Label}(M_i, q_j)),$$

where $w_i$ is the weight of the model $M_i$, $N = \sum_i w_i |J_i|$ is the normalization factor, and $\text{Label}(M_i, q_j)$ is the pre-evaluation of the answers generated by $M_i$ for the question $q_j$. In our experiments, this evaluation is automatically conducted by ChatGPT.

The $\text{Distance}$ function represents distance between the model's self evaluate and the label. In our case, we initially establish a one-to-one correspondence between evaluations and scores, as outlined in Table 2. Subsequently, we employ the absolute difference in scores as the $\text{Distance}$ function. When the model's evaluation is represented as a probability distribution across classes, the $\text{Distance}$ is computed as the mathematical expectation.

Inspired by the MAML algorithm (Finn et al., 2017) in meta-learning, we have devised a specific algorithmic workflow, as illustrated in Algorithm 1. Prior to running the algorithm, for each model, we select $u$ questions corresponding to its knowledge base along with their evaluations by the model. Additionally, we include $v$ questions randomly chosen from beyond its knowledge base, along with their evaluations. These $w = u + v$ question-evaluation pairs constitute the training dataset specific to that model $M_i$, denoted by $D_i$. In each batch of the algorithm's execution, we choose $n$ models and divide the batch into multiple mini-batches. Within each mini-batch, we randomly select $b$ data points $d_i = (d_i^{(1)}, \ldots, d_i^{(b)})$ from their respective evaluation datasets for the $n$ models, and then calculate the sum of these loss values of both $M_i$ and the base model $M_0$, denoted as $Loss = \sum_i (\text{Loss}(M_i, d_i) + \text{Loss}(M_0, d_i))$, as the overall loss. This loss is then used for gradient updates on the LoRA employed for evaluation.

In other words, for each set of data, we always compute the loss simultaneously using the base model $M_0$ and one of the fine-tuned models $M_i$. This is equivalent to assigning a weight of $1/2$ to the base model $M_0$ and assigning weights of $1/(2N)$ to each of the fine-tuned models $M_i$.

We would like to emphasize that, as language models can output logits, we are effectively using the output probabilities for various classes to calculate the Distance. Consequently, the corresponding Loss function is differentiable.

---

**Algorithm 1** Meta-cognition training algorithm

---

$\text{train\_Meta\_cognition}(E; M_i, D_i)$
**for** epochs = 1 to $V$ **do**
    Randomly group the entire set of models $M_i$ (except the base model $M_0$) into sets of $n$ models each.
    There are $B = N/n$ batches, each batch $B_j$ contains $n$ models.
    **for** j = 1 to $B$ **do**
        For every $M \in B_j$, randomly divide corresponding dataset $D$ into $w/b$ mini-batches $d_k$ of $b$ data points
        **for** k = 1 to $w/b$ **do**
            $ev_M = \text{Evaluation}(M, E, d_k)$, for every $M \in B_j$;
            Also compute $ev_{M_0} = \text{Evaluation}(M_0, E, d_k)$, for base model $M_0$;
            $\text{Loss} = \sum_{M \in B_j} (\text{Distance}(ev_M, label_M) + \text{Distance}(ev_{M_0}, label_{M_0}))$, where $label_M$ is the corresponding label, and $label_{M_0}$ is the label corresponding to the base model $M_0$;
            Back propagate through the Loss function to compute the gradient of parameters of $E$;
            Update parameters of $E$ through gradient decent.
        **end for**
    **end for**
**end for**

---

## 4.2 EXPERIMENT

In our approach, the train-test split will apply to the 160 domain specific LoRA models instead of the question-answer pairs. We randomly choose $N = 120$ LoRA models as train set, and the remaining 40 LoRA models as test set.

For each domain-specific LoRA model $\text{LoRA}_i$, we find the 100 questions related to the corresponding topic, along with 100 random questions from other topics. We then employ the evaluation process described in Section 3 to generate ChatGPT's evaluations of the answers to the 200 questions provided by ChatGLM+$\text{LoRA}_i$. We denote the dataset of those 200 question-evaluation pairs by $D_i$.

Then, we use the above algorithm to train the evaluation lora $E$. We train for $V = 3$ epochs. In each epoch, we split the train set into $B = 30$ batches, each batch contain $n = 4$ LoRA models. In each mini-bath, we will choose $b = 5$ questions in each $D_i$. Then, we will use both the base model $M_0$ (ChatGLM-6B) and finetuned model $M_i$ (ChatGLM-6B merge with $\text{LoRA}_i$) to calculate the evaluation and the loss. We will use $M_0$ (or $M_i$) plus evaluation LoRA $E$ to output. Then, all $2bn = 40$ losses will be added together to calculate the overall loss. After that, we calculate the gradient by back propagation, and use gradient decent to update parameters of $E$ (parameters not in $E$ are frozen). In practice, due to limited computational resources, we actually calculate gradients for each of the $2bn = 40$ loss functions separately and then sum the gradients, instead of directly summing the losses and computing the gradient. Based on the fundamental calculus principle that "the sum of derivatives is equal to the derivative of the sum," our approach does not introduce any errors.

After that, we test the self evaluation LoRA $E$ for the base model and the 40 LoRAs, by the corresponding datasets. The result is in Figure 1.

(a) True labels and predicted labels for original model

| (a) | model prediction | | | | |
|---|---|---|---|---|---|
| | A | B | C | D | E |
| true label A | 814 | 441 | 32 | 5 | 0 |
| B | 154 | 4568 | 245 | 11 | 2 |
| C | 29 | 483 | 943 | 1 | 1 |
| D | 5 | 72 | 25 | 112 | 0 |
| E | 0 | 21 | 10 | 0 | 26 |

(b) True labels and predicted labels for model+LoRAs

| (b) | model prediction | | | | |
|---|---|---|---|---|---|
| | A | B | C | D | E |
| true label A | 1038 | 1005 | 31 | 0 | 0 |
| B | 548 | 4148 | 148 | 4 | 0 |
| C | 31 | 365 | 531 | 3 | 0 |
| D | 4 | 49 | 11 | 57 | 0 |
| E | 0 | 10 | 2 | 0 | 15 |

Figure 1: The testing result for our evaluation model

Table 3: Evaluations and scores

| TYPE | PRECISION | RECALL | F1 SCORE | AVERAGE DISTANCE |
|---|---|---|---|---|
| original | 0.7811 | 0.6469 | 0.7077 | 0.3229 |
| with LoRA | 0.7649 | 0.5728 | 0.6551 | 0.3708 |
| Total | 0.7752 | 0.6183 | 0.6879 | 0.3468 |

### 4.3 DISCUSSION

Whether for the original model or the LoRA fine-tuned models, the resulting matrices of our results are (nearly) diagonally dominant, demonstrating that our evaluation model has indeed learned metacognitive abilities. This metacognitive ability remains preserved when combining the original model with LoRA models not used during training, from the test set. This indicates that we have achieved our goal: metacognitive abilities that are adaptable to different fine-tunings, similar to those of humans.

We further investigate the ability of our evaluation model to detect hallucinations. We'll classify C (severe hallucination) and D (nonsense) as hallucination, and A,B,E as no hallucination. Then, the precision, recall and F1 score are shown in Table 3. We also calculate the average distance of predicted class and the actual class, according to the class-score correspondence in Table 2. It appears that in LoRA fine-tuned models, there is a slight decrease in accuracy in the evaluation, but it remains acceptable.

## 5 FURTHER DISCUSSIONS

### 5.1 COMPARE WITH OTHER HALLUCINATION DETECTION METHODS

To our knowledge, our method is the only one that detects hallucinatory text by evaluating before the model's output. The model only need to output one token for detection. This feature makes our method very efficient during inference. The comparison with other methods: BARTScore(Yuan et al., 2021), GPTScore(Fu et al., 2023), Tagged prompts(Feldman et al., 2023) and SelfCheck-GPT(Manakul et al., 2023), is shown in Table 4.

### 5.2 LIMITATIONS

Our method also have some shortcomings: our method is not very accurate, since we only use the model's input prompt to detect hallucination. Also, to make our method adaptable for fine-tuned models, we have to frequently switch between models during training, which makes our training process quite slow. And, when we get an evaluation adapter, the adapter can only be adapted to the certain base model used in training, since different models have different metacognition.

We used ChatGPT for dataset generation and automatic answer evaluation, which inevitably introduced some noise, although this is unrelated to our method itself.

Table 4: Comparison of hallucination detection methods

| METHOD | INTERNAL DATA | DETECT TIMING | FINETUNE NEEDED | INFER SPEED |
|---|---|---|---|---|
| BARTScore | Yes | After output | No | Medium(one output) |
| GPTScore | Yes | After output | No | Medium(one output) |
| Tagged prompts | No | After output | Yes | Medium(one output) |
| SelfCheckGPT | No | After output | No | Slow(multiple outputs) |
| Ours | No | Before output | Yes | Fast (just one token) |

### 5.3 POTENTIAL FUTURE RESEARCH DIRECTIONS

Some commercial LLMs are continuously updating. An update is a process of finetuning, that means some new knowledge is introduced. Some of the questions that were previously refused to be answered can now be answered after fine-tuning, while conversely, some questions that could be answered before fine-tuning become refusals. This process has undoubtedly endowed large language models with some form of metacognitive ability. It's an interesting research topic that how to evaluate the metacognition of LLMs.

In our experiments, we only used LoRAs trained on single-topic datasets that are disjoint from each other. Therefore, it would be an interesting area for future research to investigate whether the performance is affected when using LoRAs trained on intersecting datasets that span multiple topics.

### REPRODUCIBILITY STATEMENT

You can reproduce this work using the code in this repository: https://github.com/LLMworker/metacognition

I did not leak any personal information in the above link.

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

APPENDIX

## A  THE FULL LIST OF THE 160 TOPICS

Here are the full list of the 160 topics used to generate the 16000 question-answer pairs. We call this dataset "ChatGPT ground truth".

| SUBJECT | TOPICS | | |
|---|---|---|---|
| Mathematics | Algebraic Geometry | Algebraic Topology | Combinatorics |
| | Partial Differential Equations | Number Theory | Category Theory |
| | Mathematical Logic | Functional Analysis | Lie Theory in Mathematics |
| | Dynamical Systems | Complex Analysis | Real Analysis |
| | Differential Geometry | Representation Theory | Probability Theory |
| | Topological Data Analysis | Abstract Algebra | Cryptography |
| | Mathematical Optimization | Mathematical Physics | |
| Physics | Condensed Matter Physics | Quantum Mechanics | Electromagnetism |
| | Thermodynamics | Particle Physics | Astrophysics |
| | Nuclear Physics | General Relativity and Cosmology | Optics and Photonics |
| | Plasma Physics | Biophysics | Computational Physics |
| | Acoustics and Vibration | Nonlinear Dynamics and Chaos | Quantum Field Theory |
| | Classical Mechanics | String Theory | Solid State Physics |
| | Geophysics | Statistical Mechanics | |
| Biology | Plant Taxonomy | Microbial Taxonomy | Vertebrate Taxonomy |
| | Paleontology | Entomology | Molecular Biology |
| | Cell Biology | Genetics | Pathology |
| | Biotechnology | Zoology | Marine Biology |
| | Neurobiology | Developmental Biology | Anatomy |
| | Evolutionary Biology | Virology | Reproductive Biology |
| | Pharmacology | Epidemiology | |
| Chemistry | Organic Chemistry | Inorganic Chemistry | Physical Chemistry |
| | Analytical Chemistry | Biochemistry | Polymer Chemistry |
| | Environmental Chemistry | Nuclear Chemistry | Materials Chemistry |
| | Medicinal Chemistry | Geochemistry | Astrochemistry |
| | Computational Chemistry | Surface Chemistry | Electrochemistry |
| | Photochemistry | Thermochemistry | Spectroscopy |
| | Crystallography | Green Chemistry | |

| SUBJECT | TOPICS | | |
|---|---|---|---|
| Engineering | Mechanical Engineering
Chemical Engineering
Industrial Engineering
Materials Engineering
Petroleum Engineering
Structural Engineering
Marine Engineering | Electrical Engineering
Aerospace Engineering
Computer Engineering
Nuclear Engineering
Agricultural Engineering
Geotechnical Engineering
Systems Engineering | Civil Engineering
Biomedical Engineering
Environmental Engineering
Telecommunications Engineering
Mining Engineering
Robotics Engineering |
| Medicine | Anesthesiology
Emergency Medicine
Geriatrics
Infectious Diseases
Obstetrics and Gynecology
Orthopedics
Psychiatry | Cardiology
Endocrinology
Hematology
Nephrology
Oncology
Otolaryngology (ENT)
Pulmonology | Dermatology
Gastroenterology
Immunology
Neurology
Ophthalmology
Pediatrics |
| Computer Science | Artificial Intelligence
Computer Vision
Human-Computer Interaction
Software Engineering
Computer Security
Bioinformatics
Game Development | Machine Learning
Natural Language Processing
Computer Networks
Databases
Computer Architecture
Web Development
Compiler Design | Data Science
Computer Graphics
Operating Systems
Complexity Theory
High Performance Computing
Mobile Development |
| Social Science | Sociology
Economics
Geography
Demography
Law
Criminology
Gender Studies | Psychology
Political Science
Linguistics
International Relations
Social Work
Communication Studies
Environmental Studies | Anthropology
History
Archaeology
Public Policy
Education
Cultural Studies |

# B  PROMPTS USED FOR AUTOMATIC EVALUATION

## B.1  PROMPT FORMAT

Here is the prompt for ChatGPT to evaluate ChatGLM-6B's answer. Note that the "ground truth" (we use the answer generated by ChatGPT itself as the ground truth, since it works perfectly when answering the questions generated by itself) part is very important. Without it, ChatGPT cannot give accurate evaluation.

```
## Question:
{The question}

## Ground truth:
{The ground truth given by ChatGPT}

## Answer by a language model:
{The answer to evaluate, given by ChatGLM-6B
(base model, or with LoRA)}

## Evaluate without explanation (A. clear answer
B. minor error C. severe hallucination D. nonsense
E. refuse to answer or other cases):
```

## B.2  EXAMPLE

Here is an example prompt.

## Question:
What is the Born rule?

## Ground truth:
The Born rule, also known as the Born probability rule or Born's statistical interpretation, is a fundamental concept in quantum mechanics. It was formulated by the German physicist Max Born in 1926 and is used to calculate the probabilities of different outcomes in quantum systems.

According to the Born rule, for a quantum system described by a wave function ($\psi$), the probability (P) of finding a particle or a system in a particular state is given by the square of the absolute value of the wave function amplitude at that state. In mathematical terms, it can be expressed as follows:
$P = |\psi|^2$
Here, $|\psi|^2$ represents the magnitude squared of the wave function ($\psi$), and it provides the probability density for finding the particle at a specific position or in a specific state. The Born rule is crucial in quantum mechanics as it allows us to predict the probabilities of different measurement outcomes and understand the probabilistic nature of quantum systems.

It's important to note that while the Born rule provides probabilities for the outcomes of measurements, it does not determine which outcome will be observed in any particular experiment. Instead, it helps us understand the statistical distribution of measurement results when dealing with quantum entities such as particles or systems. The actual outcome of a measurement is subject to the inherent randomness and uncertainty inherent in quantum mechanics.

## Answer by a language model:
The Born rule is a mathematical formula that describes the probability of an event occurring. It is named after Max Born, a German physicist who first introduced this formula in the 1920s. The Born rule states that, given the opportunity to choose between two possible events, the event with the higher probability of occurring will be chosen.
Here is the Born rule in a simple form:
$P(A|B) = (P(B|A) * P(A))/P(B)$
where $P(A|B)$ is the probability of event A occurring given that event B has occurred, $P(B|A)$ is the probability of event B occurring given that event A has occurred, P(A) is the probability of event A occurring, and P(B) is the overall probability of event B occurring.

## Evaluate without explanation (A. clear answer B. minor error C. severe hallucination D. nonsense E. refuse to answer or other cases):

Output: C. severe hallucination

If we delete the "ground truth" section, ChatGPT will fail to detect the hallucination.

## Question:
What is the Born rule?

## Answer by a language model:
The Born rule is a mathematical formula that describes the probability of an event occurring. It is named after Max Born, a German physicist who first introduced this formula in the 1920s. The Born rule states that, given the opportunity to choose between two possible events, the event with the higher probability of occurring will be chosen.
Here is the Born rule in a simple form:
$P(A|B) = (P(B|A) * P(A))/P(B)$
where $P(A|B)$ is the probability of event A occurring given that event B has occurred, $P(B|A)$ is the probability of event B occurring given that event A has occurred, P(A) is the probability of event A occurring, and P(B) is the overall probability of event B occurring.

## Evaluate without explanation (A. clear answer B. minor error C. severe hallucination D. nonsense E. refuse to answer or other cases):

Output: B. minor error

Note that we've edit the original prompt a bit for latex formatting issues.

