# OpenReview forum: "Teach Large Language Models the Concept of Meta-cognition to Reduce Hallucination Text Generation"
_ICLR.cc/2024/Conference — Submitted to ICLR 2024_

### Official Review · Reviewer_KFBC · 2023-10-31

**Soundness:** 1 poor
**Presentation:** 2 fair
**Contribution:** 1 poor
**Rating:** 3
**Confidence:** 4

**Summary:**

The paper introduces the concept of "metacognition", which refers to the model's ability to accurately self-evaluate one's own capabilities before generating text. They then propose to equip language models with this ability by first using ChatGPT to generate factual question-answer pairs in different domains, and then finetuning ChatGLM with LoRA in a meta-learning style. They claim that their model learns metacognitive abilities after training.

**Strengths:**

1. The paper constructs a factual question-answering dataset that can be helpful for future research.

**Weaknesses:**

1. One major weakness of this paper is that the concept of "metacognition" is not formally defined. In the introduction part, it mentions that it is the model's ability accurately self-evaluate one's own capabilities before generating text. However, no formal definition is presented and it is unclear how we can quantify such abilities. Therefore, it is unclear to how to evaluate a model's metacognition abilities and the experiments cannot support their claims such as "their models can learn metacognitive abilities after training."
2. The paper proposes several hypotheses that may be ungrounded. For example, in the introduction part they claim that "RLHF is the main source of hallucination of recent LLMs", but MLE-trained models can also hallucinate and RLHF can help with reducing hallucinations [1]. There are also other kinds of similar claims in the introduction section such as "LLMs are not enough to fit the world’s model."
3. The writing of the paper lacks coherence. For example, they have put a lot of efforts on writing the retrieval-augmented models but didn't compare or use them. Also, it is unclear how one can reframe the task of "instilling in LLMs the discretion to search only when absolutely necessary" to "teach LLMs to accurately self-evaluate their own capabilities before generating text." Teaching LLMs to self-evaluate their own capabilities is one way to implement this task, but they are not interchangeable.
4. There are related works on model calibration and retrieval-augmented models that they didn't discuss in depth.



[1]  Sun et al., Aligning Large Multimodal Models with Factually Augmented RLHF.

**Questions:**

1. How do you quantify "meta-cognition"?
2. What is the difference between "meta-cognition" and "model calibration"?

---

> ### Author Response · Authors · 2023-11-16
>
> Thank you for your thoughtful review and valuable suggestions. The preparation time for this paper was indeed relatively tight. I will carefully refine it in accordance with your advice.
>
> Regarding the statement "RLHF is the main source of hallucination of recent LLMs," I want to clarify that I am referring to the RLHF performed by mainstream AI research institutes. In order to make Language Models (LLMs) useful, the reward model is more likely to favor a formally good answer rather than rejecting an answer. I think that may lead hallucinations.  These intuitive guesses indeed lack theoretical and experimental evidence. You rightly point out that if the reward model is factually augmented, the results could be substantially different. I appreciate you bringing this to my attention and providing such insightful information.

---

### Official Review · Reviewer_z9Ch · 2023-10-31

**Soundness:** 2 fair
**Presentation:** 1 poor
**Contribution:** 2 fair
**Rating:** 1
**Confidence:** 4

**Summary:**

The authors introduce an algorithm designed to equip language models with sustained meta-cognitive abilities, drawing inspiration from meta-learning principles. The paper introduces an algorithm designed to endow large language models with metacognitive abilities, allowing them to evaluate their own capacity to generate responses to human instructions, and as a result, curtail the production of responses when the model is not adequately equipped to answer, significantly diminishing the incidence of hallucinatory text generation. Additionally, the authors present a novel ground truth dataset, encompassing 16,000 questions spanning 160 meticulously categorized domains, which can be utilized for fine-tuning and evaluating large language models.

**Strengths:**

This submission set eyes on an interesting topic of introducing meta-learning to LMs so as to address the issue of hallucination. Their efforts in putting together a manuscript for this submission are acknowledged and appreciated.

**Weaknesses:**

(1) The manuscript’s clarity and structural integrity could be greatly improved. The format of the manuscript leans more towards a school report than a scholarly research paper, and it currently presents challenges in understanding the main contributions and claims:
* The motivation and main goal of the study need clearer definition; it’s uncertain whether the focus is on addressing hallucination in text generation or applying meta-learning to LLMs.
* The transition from discussing hallucination in Section 2 to model selection and metacognitive capabilities in Section 3 is abrupt and confusing. A more systematic introduction to the problem, followed by the detailed methodology, would enhance comprehension.




(2) The design and execution of the experiments require substantial revision to robustly support the authors’ claims:
* Clarification is needed on the specific text generation task under study. Text generation is a very broad domain with many subtasks. The authors mention summarization and dialogue generation in the related work, but later train the model with QA setup. What exactly is the task that this work is trying to study?
* Why did the authors choose ChatGLM-6B? Is it because this model is bilingual? Then why didn’t the authors compare to other multilingual models such as BLOOM? Is it because this model supports multi-turn dialogue? Then there are also other dialogue models/chatbots available. Why didn’t the authors compare to any of these related baselines?
* The evaluation metrics need a detailed description, and the results presented in the tables require thorough analysis and interpretation.
* The authors mentioned that there are a number of benchmarks for evaluating hallucination in text generation, then why didn’t the authors report results on any of these benchmarks?



(3) This submission did not mention a number of related works throughout the manuscript:
* In the introduction, the authors briefly mentioned that metacognition is a concept in psychology, but did not point to any related studies.
* The authors mention that their “algorithm draws inspiration from meta-learning”, but did not discuss any previous work on meta-learning.


(4) Typos, formatting issues, etc:
* Unpaired quotation marks: (e.g., Section 1: ”hallucination”);
* Missing reference when mentioning GPT-3, PaLM, Llama, etc in Section 3.1;
* Misused citation format: e.g., Section 2: “Another approach is shown in (Feldman et al., 2023)” -> the parentheses around the citation are unnecessary and should be removed to adhere to standard citation practices;
* Table 5 and tables in the appendix extend beyond the page width;
* What makes Figure 1 a figure? It appears to be two tables to me.

**Questions:**

Please kindly refer to the questions raised in the Weaknesses section.

---

> ### Author Response · Authors · 2023-11-16
>
> Thank you for carefully reading my work and providing valuable feedback. I acknowledge that the preparation time for this paper was relatively tight, leading to some formatting errors. Additionally, I recognize that the scale and types of experiments were somewhat limited, and the analysis of experimental data could be more robust. I sincerely apologize for these shortcomings.
>
> In the upcoming revision, I will diligently rectify the formatting errors and strive to enhance both the scale and diversity of my experiments. I also aim to strengthen the analysis of experimental data. Your guidance is greatly appreciated, as it will contribute to the improvement and overall quality of the paper.
>
> Once again, thank you for your review and valuable suggestions.

---

### Official Review · Reviewer_AYMX · 2023-11-01

**Soundness:** 1 poor
**Presentation:** 2 fair
**Contribution:** 3 good
**Rating:** 3
**Confidence:** 4

**Summary:**

This paper introduces an algorithm that endows language models with persistent metacognitive capabilities.  This empowers these models to evaluate their competence when interpreting human instructions, thereby avoiding the generation of responses beyond their abilities and mitigating the production of hallucinatory text.

**Strengths:**

The paper presents a highly innovative concept, utilizing the idea of meta-learning to address hallucinations. It is also the first method to mitigate hallucinations before the model's output. In terms of writing, the introduction of hallucination-related terms at the beginning of the article uses numerous vivid metaphors, making it very easy to understand and read.

**Weaknesses:**

1、This article seriously lacks quantitative experimental results to verify the effectiveness of the method. The claim of the entire paper does not have sufficient experimental support.

2、However， there is also a lack of theoretical discussion further validating the method's effectiveness.

2、Additionally, due to the need for a substantial number of fine-tuned models, practical application can be challenging.

**Questions:**

The phrase "Reduce Hallucination Text Generation" in the title seems awkward. A more appropriate expression  could be "Reducing Hallucinations in Text Generation" or "Reducing Hallucinatory Text Generation".

---

> ### Author Response · Authors · 2023-11-16
>
> Thank you for reviewing and giving good advices. The preparation time for this paper was relatively tight, leading to a lack of experiments and theoretical discussions. I'll refine it later.
>
> May all go well with you!

---

### Official Review · Reviewer_KoVi · 2023-11-02

**Soundness:** 2 fair
**Presentation:** 3 good
**Contribution:** 2 fair
**Rating:** 6
**Confidence:** 3

**Summary:**

This paper introduces an algorithm designed to equip language models with enduring meta-cognitive capabilities. Drawing inspiration from meta-learning, the approach involves a process of fine-tuning models on diverse datasets, including the original untuned base model.

what contributions does it make:
1.This approach allows models to self-assess their competence when interpreting human instructions.
2.This method improves the model's response quality by preventing responses beyond their learned capacities and reducing the generation of misleading or hallucinatory text.
3.The paper provides a new ground truth dataset for evaluating large language models.

**Strengths:**

1.The paper is written in an easy-to-understand manner.
2.The meta-cognitive ability adapts to the various fine-tuned versions of the primary model. This adaptability offers evaluations aligned with the knowledge capacity of the fine-tuned models.
3.In each training cycle, the algorithm randomly selects multiple fine-tuned model versions and evaluates their meta-cognitive abilities.

**Weaknesses:**

1.The experiment conducted was inadequate, relying solely on ChatGLM-6B, which possesses a limited knowledge base.
2.Table 1 lacks any analysis or interpretation.

**Questions:**

1.In the dataset constructed in this paper, the questions must be in line with objective facts, and they are all in English, can this be extended to a more general situation? How well does the model generalize outside of these specific scenarios?
2.Compared with MAML, what do the support data and query data correspond to in the algorithm proposed in this paper?

---

> ### Author Response · Authors · 2023-11-16
>
> Thank you for reviewing and giving good questions and advices.

---

### Meta-Review · Area_Chair_PAsj · 2023-12-05

**Metareview:**

The paper introduces the concept of "meta-cognition" in the context of language models, aiming to equip them with the ability to self-assess their capacity to generate accurate responses to human instructions. However, the paper faces several weaknesses, including the lack of a formal definition and quantification of "meta-cognition," unclear claims in the introduction, and a lack of coherence in the writing. Additionally, the experimental support for the proposed method is limited, and related work on model calibration and retrieval-augmented models is not discussed in depth. Overall, while the concept of enhancing meta-cognitive abilities in language models is promising, the paper needs significant refinement and more robust experimental validation to substantiate its claims.

**Justification For Why Not Higher Score:**

The paper's current form is unfortunately below the acceptance threshold of ICLR.

**Justification For Why Not Lower Score:**

N/A.

---

### Decision · Program_Chairs · 2024-01-16

Reject